# Occurrence of *Giardia duodenalis* in Cats from Queretaro and the Risk to Public Health

**DOI:** 10.3390/ani13061098

**Published:** 2023-03-20

**Authors:** Nerina P. Veyna-Salazar, Germinal J. Cantó-Alarcón, Andrea M. Olvera-Ramírez, Felipe J. Ruiz-López, Rodolfo Bernal-Reynaga, Isabel Bárcenas-Reyes, Marina Durán-Aguilar

**Affiliations:** 1Doctorado en Ciencias Biológicas, Facultad de Ciencias Naturales, Universidad Autónoma de Querétaro, Avenida de las Ciencias S/N Juriquilla, Delegación Santa Rosa Jáuregui, Querétaro 76230, Mexico; 2Cuerpo Académico Mejoramiento Animal Integral, Facultad de Ciencias Naturales, Universidad Autónoma de Querétaro, Avenida de las Ciencias S/N Juriquilla, Delegación Santa Rosa Jáuregui, Querétaro 76230, Mexico; 3Cuerpo Académico Salud Animal y Microbiología Ambiental, Facultad de Ciencias Naturales, Universidad Autónoma de Querétaro, Avenida de las Ciencias S/N Juriquilla, Delegación Santa Rosa Jauregui, Querétaro 76230, Mexico; 4Centro Nacional de Investigación en Fisiología y Mejoramiento Animal, INIFAP-SAGARPA, México. Km1. Carr. Ajuchitlán-Colón Ajuchitlán, Querétaro 76280, Mexico; 5Cuerpo Académico de Salud Pública, Unidad de Investigaciones en Salud Pública “Dra. Kaethe Willms”, Facultad de Ciencias Químico Biológicas, Universidad Autónoma de Sinaloa, Ave. de las Américas y Blvd. Universitarios, Ciudad Universitaria, Culiacán 80100, Mexico

**Keywords:** *Giardia*, zoonoses, assemblages, cats, risk, Mexico

## Abstract

**Simple Summary:**

*Giardia duodenalis* is a flagellated protozoan that has been reported worldwide. Its resistance to adverse climates and its wide host range makes this parasite a serious problem specifically in developing countries where personal hygiene practices are inadequate. The magnitude of the risk of transmission from cats to humans is not well known; to date, in Mexico, there has not been a single study of the presence of *G. duodenalis* in cats. Therefore, the objective of this work is to determine the frequency and importance of the cat as a potential zoonotic reservoir of *Giardia*. Sampling of feces was performed in private clinics and an animal control unit. A direct microscopy diagnosis was carried out, and once positive results were obtained, PCR and RFLP were performed in order to obtain assemblages. The results obtained in the present study, in terms of the genetic characterization, showed only assemblage A, which indicates that the cat is an important source of transmission to humans, with consequences for public health.

**Abstract:**

*Giardia* is a protozoan that affects humans as well as a wide range of domestic species. It is distributed worldwide, and the highest frequency is seen in developing countries. Due to the potential for domestic cats to be carriers of this parasite and subsequently transmit the infection to humans, it is important to know the risk of transmission. For this reason, the objective of this study was to determine the frequency of this parasite in the cat population of the city of Santiago de Queretaro, Mexico, and identify the assemblages present to determine the role this host plays in public health, this being the first study of its type to be performed in the country. This was a cross-sectional study during which 200 fecal samples were collected from cats of both sexes and varying ages and strata of origin. The samples were analyzed by microscopy following the flotation technique, having obtained a general frequency of 25%. *Giardia* cysts were found at higher frequency in pasty stools. The assemblages found were zoonotic, specifically assemblage A, which suggests that the cat poses an important risk for the dissemination of the parasite to humans, making it an important public health problem.

## 1. Introduction

*Giardia duodenalis* is one of the most common gastrointestinal parasites that affects humans. Its resistance to adverse climates and its wide host range make it a serious problem, more so in places that lack good personal hygiene practices [1,2]. Because giardiosis can occur following the ingestion of only 10 cysts, this protozoan is considered a risk to public health [3]. Despite having been discovered 400 years ago, the disease is considered a re-emerging disease [4]. Worldwide, about 200 million people are reported to be infected with the parasite, mainly in Asia, Africa, and Latin America, and approximately 500,000 new cases are reported annually [5].

*Giardia* is transmitted through the fecal–oral route, frequently due to the ingestion of food and water contaminated with the parasite’s cysts, or by host-to-host contact [6], The most susceptible section of a population includes infants in daycare, daycare workers, travelers to regions where the protozoan is endemic, people with immunodeficiencies and cystic fibrosis, and those who participate in oral and anal sexual practices [7].

*Giardia duodenalis* has a direct life cycle that is composed of two morphologically different stages. On the one hand, the infective stage, or cyst, is responsible for causing the infection. The cyst can survive in the environment for months under optimal conditions (4 °C) [8]. On the other hand, the invasive stage, or trophozoite, is responsible for causing the disease due to its colonization in the upper parts of the small intestine. Previously, the cyst was considered to be a cryptobiotic form; however, it was found that its oxygen consumption is 15% of that of trophozoites and it is capable of passing through the stomach and excysting in the duodenum [9].

Giardiosis can manifest symptomatically and asymptomatically; the fact that this parasite is found to a greater extent in asymptomatic patients, including cats, increases the risk of transmission to humans [10]. Although it is said that the presence of clinical signs could be linked to host factors, such as immune response, it may also be due to virulence factors of the strain of the infecting parasite [11]. However, the risk factors that may trigger the clinical disease are not yet known with certainty [12]. The most common signs of giardiosis include diarrhea, steatorrhea, abdominal pain, and weight loss [11]. Clinical manifestation of giardiosis may also be related to the assemblages present, with assemblage A being associated with acute disease and the presence of explosive diarrhea, and assemblage B being associated with chronic disease [4,13,14].

Studies indicate that cysts are passed through the stool intermittently, making diagnosis difficult and mistaking it with other gastrointestinal diseases. The low sensitivity of coproparasitoscopic tests makes detection of the disease even more difficult, for which a minimum of three serial tests are needed to observe the morphological forms [15].

The presence of *G. duodenalis* depends on variables such as geographic area, sanitary conditions, and number of animals sharing the same space. Epidemiological studies allow us to understand the importance of giardiosis in different regions of the world, characterizing the presence of the zoonotic and non-zoonotic assemblages and the risk of people living with companion animals, especially cats [16,17].

The reported prevalence of *Giardia* in humans in Mexico is variable, ranging from 2 to 50%, depending on the region, with preschool-age children being the most susceptible [18,19]. With regard to cats, studies have reported prevalences ranging from 1 to 43% (Table 1).

*Giardia* affects a wide range of species, and the fact that it can infect humans and animals raises concern regarding the risk to public health posed by companion animals [32]. The level of risk depends on the prevalence and assemblages present, and since cats are reservoirs of this parasite, this causes public health problems, affecting the health of humans and animals [33].

To date, the magnitude of the risk of transmission from cats to humans is not well known. Studies on the genetic variability of *G. duodenalis* have identified four main genetic groups. Groups A and B generally affect humans, while groups C and D affect dogs, and group F is more specific to cats. However, there are reports of groups A and B infecting cats and dogs. Group A includes four subgroups: AI to AIV. Subgroups AI and AII can be found in both humans and animals, but AIII and AIV are exclusive to animals [34,35]. Currently, mixed infections with assemblages belonging to humans and cats have been reported; therefore, from an epidemiological point of view, this suggests that there is a potential environmental reservoir for giardiosis in urban areas [36]. In Mexico, studies have been performed regarding the prevalence and presence of zoonotic assemblages in dogs, sheep, and cattle [19,37,38]; however, there are no reports of the occurrence of *G. duodenalis* or its assemblages in cats. For this reason, the present study aims to determine the occurrence of assemblage A and its sub-assemblages (AI, AII) in cats that affect humans as well as the role that cats have as potential transmitters of this protozoan to humans.

## 2. Materials and Methods

### 2.1. Study Setting and Sampling

The study was performed in the city of Santiago de Queretaro, located in the center of Mexico. This city has a territorial extension pf 263 km^2^, a population of 1,049,777, and a temperate, semi-arid climate (20 °C annual average temperature, 720 mm^2^ per year, 38.2% humidity). For the study, 200 fecal samples were obtained from cats from two different strata of origin: (1) without an owner or “unowned” (stray and feral cats), from the municipal animal control unit (UCAM), and (2) with an owner (pet cats), from private veterinary clinics. The samples were labeled and data regarding age and sex of the cats and consistency of the feces was recorded. The samples were then transported to the parasitology lab of the Autonomous University of Queretaro (UAQ), where they were stored at 4 °C until further processing.

A total of 200 samples were distributed, as follows: by strata of origin, 103 were from cats with owners and 97 from cats without owners; by age, 72 were from cats under 6 months and 128 were from cats over 6 months; by gender, 109 were from females and 91 were from males; and by stool consistency, 139 stools were firm and 61 were pasty.

### 2.2. Coproparasitoscopic Analysis

Two grams of each fecal sample were processed using the flotation technique with zinc sulfate, as described by Dryden et al. [39]. A drop of fecal suspension was transferred to a microscope slide with a cover slip and examined at 40X magnification for identification of cysts. A sample was considered positive if cysts were observed.

### 2.3. Sample Pooling

After identifying the positive samples, seven pools were made. It was not possible to make more pools because of the low volume of the samples obtained from the cats. All of the positive samples with a sufficient quantity of feces were pooled. Each pool consisted of four positive samples (totaling 28 samples) in order to obtain a minimum of 5 g of feces, which is required for cyst concentration with the sucrose gradient method [40]. This technique was required due to the necessity of obtaining a sufficient number of cysts to perform an optimal DNA extraction [41].

### 2.4. DNA Extraction

DNA extraction was performed following the technique described by Babaei et al. [42], with some modifications to obtain more DNA from each sample. A 200 µL volume of the concentrated cyst suspension was transferred into a 2 mL Eppendorf tube with 200 µL of glass beads (0.1 mm) and 500 µL of a buffer lysis solution (100 mM NaCl, 50 mM Tris HCl, 100 mM EDTA, 1% SDS, pH 7.4) and homogenized in a Powerlyzer 24 (MO BIO, Carlsbad, CA, USA) with five 2 min cycles in liquid nitrogen and boiling water. A 40 µL volume of proteinase K and 10 µL of SDS (1 M) were added, and the reaction was incubated at 55 °C for 4 h. Subsequently, DNA was extracted using the CTAB method, as described by De Almeida et al. [43].

### 2.5. PCR, RFLP, and Sequencing

Fragments of the *β-giardin* gene were amplified in two phases using the following primers: G7 5′-AAGCCCGACGACCTCACCCGCAGTGC-3′ and G759 5′-GAGGCCGCCCTGGATCTTCGAGACGAC-3′ for an initial reaction, and G376 5′-CATAACGAC-GCCATCGCGGCTCTCAGGAA-3′ and G759 for a second reaction. The cycling conditions were those described by Caccio et al. [44]. The reaction mixture consisted of 6.25 µL of GoTaq Green Master Mix 2X (Promega, Madison, WI, USA), 0.4 µL of each primer (10 µM), 1 µL of BSA, 2.45 µL of nuclease-free water, and 2 µL of DNA (100 ng/µL). PCR products were analyzed by 2% agarose gel electrophoresis (Thermo Scientific Waltham, MA, USA), submerged in TAE 1X buffer (55 min, 75 volts), including a molecular weight marker of 100 pb (Promega, Madison, WI, USA), and blue/orange 6X loading buffer (Promega, Madison, WI, USA).

For genetic characterization, restriction fragment length polymorphism (RFLP) analysis was performed using 10 µL of the second PCR product (380 bp) and 0.25 µL of restriction enzyme HhaI (Promega, Madison, WI, USA), for a total reaction volume of 10.25 µL, and then incubated in a water bath at 37 °C for 2 h [44]. Banding patterns were analyzed by 3% agarose gel electrophoresis (Thermo Scientific Waltham, MA, USA), submerged in TAE 1X buffer (90 min, 70 volts), including a molecular weight marker of 100 pb (Promega, Madison, WI, USA), and blue/orange 6X loading buffer (Promega, Madison, WI, USA).

To validate the resulting amplicon as *Giardia duodenalis*, the second PCR product was sequenced, also using the G759 and G376 primers mentioned above. Sanger sequencing was performed by the National Laboratory of Genomics for Biodiversity (LANGEBIO-CINVESTAB, Irapuato, Guanajuato, Mexico) using a [PacBio/Illumina/Ion Torrent/etc instrument]. For validation, the sequences were compared against *G. duodenalis* reference sequences found in NCBI using the BLAST (blastn) web tool, which is available at https://blast.ncbi.nlm.nih.gov/Blast.cgi. Accesed on 5 May 2022.

### 2.6. Statistical Analysis

A χ^2^ analysis was used to determine associations between the presence of the parasite and any of the variables (age, sex, strata of origin, and stool consistency) using SPSS Statistics software, version 25.0 (IBM Corp, Armonk, NY, USA). Associations were considered statistically significant if *p* < 0.05.

## 3. Results

### 3.1. Coproparasitoscopic Analysis

Based on the zinc sulfate flotation technique, cysts were detected by microscopy in 50 out of the 200 samples analyzed, obtaining an overall occurrence of 25%. Significant statistical associations for the presence of the parasite were determined for age and stool consistency (Table 2). Detection of *G. duodenalis* was similar in cats with and without an owner (*p* > 0.05). Regarding age, the occurrence was higher in young cats (33%) than in adults (20%) (*p* < 0.05). For stool consistency, the occurrence was higher in cats with pasty stool (44%) as opposed to those with firm stool (16%) (*p* < 0.05). Finally, sex didn’t show significant association, as the occurrence was similar in males (24%) and females (25%) (*p* > 0.05).

### 3.2. PCR, RFLP, and Sequencing

Seven groups of pooled samples were obtained based on age, sex, and strata of origin. These pools showed a good number of cysts (at least 10 cysts per field with a 40x objective). A fragment specific to the *β-giardin* gene was amplified from all of them, obtaining a 753 bp product for the first PCR reaction and a 384 bp product for the second (Figure 1). Original figures were shown in Appendix A.

RFLP on the fragments corresponding to the second PCR product (384 bp) resulted in the detection of assemblage A in all the pools. Three were further determined as the AI assemblage, with a fragment pattern of 70, 100, and 190 bp, and four were assemblage AII, consistent with a fragment pattern of 70 and 210 bp (Figure 2).

Two sequences were recovered from the sanger sequencing results, one from the G759 primer and one from G376 (Appendix B). BLAST results showed 99% identity similarity to the *β-giardin* gene sequences reported in GenBank. These sequences were submitted to the GenBank: PRJNA945414.

## 4. Discussion

The estimated occurrence of *G. duodenalis* obtained in this study and the assemblages detected indicate that cats represent an important risk for the transmission of this protozoan to humans in Santiago de Queretaro, this being the first study of its type carried out in Mexico. The overall occurrence was 25%, similar to that obtained by studies on cats from Serbia and Colombia [28,45]; however, higher than that reported worldwide (12%) [46] and lower than that reported in Brazil (32%) [47]. The diagnostic method used in the present study was zinc sulfate with centrifugation, which has proven to have a sensitivity of over 72% if a single sample is analyzed [48,49]. This technique has been considered the gold standard for the diagnosis of *Giardia* in dogs and cats [39]. However, due to the intermittent excretion of cysts and because only one sample was collected rather than the three recommended [49] the occurrence obtained in this study could be underestimated.

Differences in occurrence were seen in each of the categories defined, such as strata of origin (owned vs. unowned), age, sex, and stool consistency. Regarding their origin, the occurrence was slightly higher for cats with owners (28%) than for cats without owners (21%); in Mexico, it is more common for household cats to have access to the exterior and roam the streets, which inadvertently puts their owners at higher risk for infection [50].

A significant association with age was identified in this study, where young cats (<6 months) had a higher occurrence (30%) than adults (21%) (*p* < 0.05). Similar findings were reported by Nikolic et al. [45], in which *G. duodenalis* was detected in young cats at a frequency of 30.4% but only in 19% of adults. Therefore, the age of the animals could potentially be a risk factor for becoming infected with this parasite, most likely due to their adaptive immunity not being fully developed [45]. Consequently, these young cats can excrete high amounts of cysts per gram of feces; therefore, they represent a significant source of infection [45,51,52]. In contrast, sex was not a statistically significant factor (*p* > 0.05), obtaining a frequency of 24% for males and 25% for females, coinciding with previously published studies [22,53].

As our results show, stool consistency was an important factor for detecting giardiosis in cats, as diarrhea or pasty stool with steatorrhea are among the main clinical signs observed in cats with this disease [2]. Therefore, in the present study, cats with pasty stools (44%) were significantly (*p* < 0.05) more likely to be infected than those that had firm stools (16%) (Table 2). These findings agree with those of other studies in which *G. duodenalis* was detected more frequently from animals with diarrhea [53,54,55].

For the molecular analysis, we pooled the positive samples to ensure a high yield of DNA, and this strategy has proven to be cost-effective and efficient for obtaining precise results [56]. Pooling of samples has also been commonly used in veterinary medicine as a fast and safe method for estimating prevalene based on microscopy results [57,58]. Additionally, molecular studies have used this pooling method to increase sensitivity at the group level, which has shown to be a reliable method for the detection of several microorganisms [46,59,60]. Here, PCR was performed on each of the seven pools targeting a fragment of the *β-giardin* gene due to its specificity for *G. duodenalis* and its high sensitivity for genetic characterization, both of which were essential for the objectives of this study [38,44,61].

The results show assemblage A, which suggests that the cat could be an important transmitter of this protozoan to humans [24] In this study, assemblage B was not analyzed due to the fact that its presence in Mexico is very low and it has only been found twice in humans and never in domestic animals [19,62]. The presence of assemblage A raises the concern of zoonotic transmission due to the association with the presence of clinical signs of the disease [13]. This coincides with other reports in which the zoonotic assemblages dominated in the cat population [24,63,64,65]; in contrast, some studies have detected cat-specific assemblage (F) at a higher frequency [66,67,68]. Here, further genetic characterization showed that assemblage AII was present at a higher proportion (57%) than assemblage AI (42%), coinciding with Procesi et al. [63]. Because AII is predominantly present in humans and is associated with virulence factors that may trigger clinical manifestations, this finding raises the concern for zoonotic transmission [69,70]. Studies carried out in North America have reported three assemblages in cats: AI and AII from group A and groups B and F [71,72], coinciding also with assemblage A detected here. In Mexico, the current cat population is at approximately 16.2 million; pet cats are not regularly spayed/neutered and usually have unrestricted access to the outdoors. Also, some can become free-roaming cats, without necessary preventative care to control these diseases. Gerold and Jessup have described that free-roaming cats represent a significant source of zoonotic diseases, including rabies, toxoplasmosis, larva migrans, tularemia, and plague [33]. Consequently, the high level of interaction among cats can lead to an increased risk of cat-to-cat transmission of pathogens, with subsequent transmission to humans and other animals.

The close contact that exists between humans and their pet cats needs to be taken into consideration due to the high occurrence and assemblages observed [73]. From this perspective, it is important to emphasize the need to develop effective strategies for the control of these zoonotic diseases, as well as raise awareness among the population and encourage responsible ownership and proper management in order to reduce the risk of transmission.

## 5. Conclusions

The high occurrence rate of *G. duodenalis* observed in this study, as well as the presence of assemblages A only, makes evident the source of infection that cats represent for humans and other domestic animals. With this knowledge, control measures need to be carried out in order to minimize or eliminate the risk of *Giardia* transmission to cat owners.

## Figures and Tables

**Figure 1 animals-13-01098-f001:**
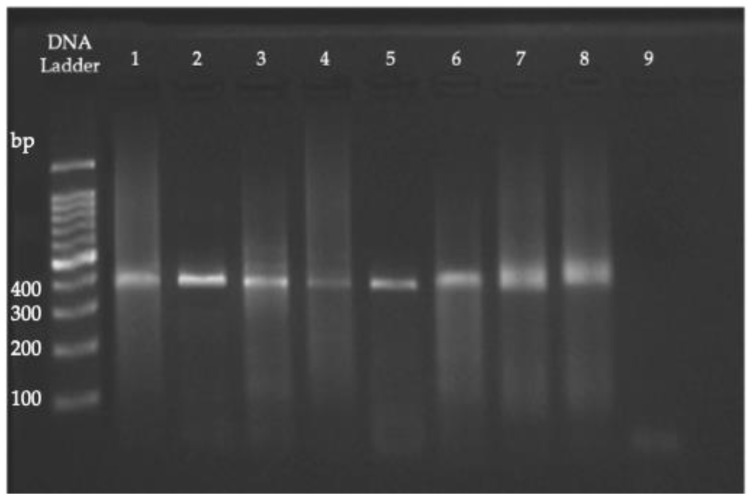
Amplicons of 384 bp in size from the second PCR reaction (*G. duodenalis*). Lane 1 corresponds to control; lanes 2–8 correspond to the pools analyzed; lane 9 corresponds to the negative control.

**Figure 2 animals-13-01098-f002:**
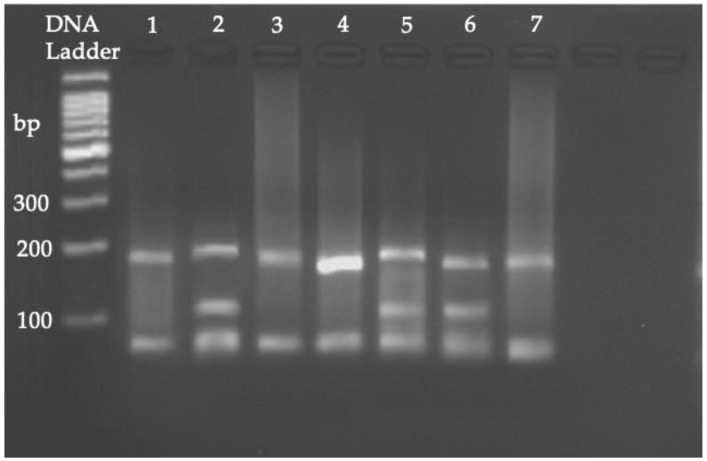
RFLP analysis of a 384 bp fragment of the *β-giardin* gene (*G. duodenalis*). Lanes 2, 5, and 6 correspond to the AI assemblage (190, 100 and 70 bp) and lanes 1, 3, 4, and 7 correspond to the AII assemblage (210 and 70 bp).

**Table 1 animals-13-01098-t001:** Occurrence, origin, age, and assemblages of *Giardia duodenalis* in cats.

Location	Age and Origin	Total	Occurrence (%)	Assemblage	Reference
Germany	Owned, stray, Shelter (Adults and kittens)	145	17.9%	D	2018 [20]
Italy	Owned, Stray (Kittens, Adults)	181	6.1%	A, F	2011 [21]
Romania	Owned (Kittens, Adults)	181	28%	Not reported	2011 [22]
Poland	Owned (No age)	160	3.75%	A, B, D	2011 [23]
USA	Owned (No age)	250	13%	AI, AII, F	2007 [24]
USA	Owned (Kittens, Adults)	211,105	0.58%	Not reported	2006 [25]
Japan	Kennels, Owned (Kittens, Adults)	600	40%	Not reported	2005 [26]
Colombia	Stray (Kittens, Adults)	46	6.5%	F	2006 [27]
Colombia	Owned (Kittens, Adults)	203	20%	Not reported	2019 [28]
Brazil	Owned, shelters (No age)	19	Not reported	AI, F	2007 [29]
Chile	Owned (No age)	230	19%	Not reported	2006 [30]
Costa Rica	Owned (No age)	9	57.1%	Not reported	2011 [31]

**Table 2 animals-13-01098-t002:** Distribution of samples and statistical association analysis by stratum, age, stool consistency, and gender.

	Positive	Negative	Occurrence (%)	*p* Value	Odds Ratio	95% CI
Total	50	150	25%			
Strata of origin						
Owned	29	74	28	0.28827	1.41	0.731–2.7
Unowned	21	76	21			
Age						
<6 Months	24	48	33%	0.041227	1.96	1.02–3.76
>6 Months	26	102	20%			
Stool consistency						
Pasty	27	34	44%	0.000031	4.00	2.03–7.8
Firm	23	116	16%			
Sex						
Male	22	69	24%	0.805723	0.92	0.48–1.75
Female	28	81	25%			

## Data Availability

Not applicable.

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
