# Peer review of "Occurrence of Giardia duodenalis in Cats from Queretaro and the Risk to Public Health"

_animals, 2023, doi:10.3390/ani13061098_

Round 1

Reviewer 1 Report

I have read with interest the manuscript “Prevalence of Giardia intestinalis in cats from Queretaro and the risk to public health”

However, several topics should be answered by the authors.

In Material and Methods section

-The presence of parasites was determined by concentrating cysts from feces, followed by conventional microscopy but this technique has a poor sensitivity, so possibly the positive samples are underestimated

Immunofluorescence assays (IFAs) can be used to stain fecal smears without initial concentration and has more sensitivity   than concentration/microscopy combination.

I recommend reanalyzing the samples using IFA, previously to PCR assays.

Gotfred-Rasmussen H, Lund M, Enemark HL, Erlandsen M, Petersen E. Comparison of sensitivity and specificity of 4 methods for detection of Giardia duodenalis in feces: immunofluorescence and PCR are superior to microscopy of concentrated iodine-stained samples. Diagn Microbiol Infect Dis. 2016 Mar;84(3):187-90. doi: 10.1016/j.diagmicrobio.2015.11.005. Epub 2015 Nov 10. PMID: 26707069.

In results section

- Where obtained the positive control?

- Authors should include controls for Assemblages A, B and F in PCR, at least. It is not clear to the reader that the samples are Assemblages A.

Author Response

First of all, we are very thankful to the reviewer 1 for the time he/she took to review this paper and for the very valuable comments

I have read with interest the manuscript “Prevalence of Giardia intestinalis in cats from Queretaro and the risk to public health”

However, several topics should be answered by the authors.

In Material and Methods section

-The presence of parasites was determined by concentrating cysts from feces, followed by conventional microscopy but this technique has a poor sensitivity, so possibly the positive samples are underestimated

We agree with the reviewer that the occurrence of Giardia might be higher and that was discussed on lines 396-401 as follows “The diagnostic method used in the present study was the zinc sulfate with centrifugation, which has proven to have a sensitivity of over 72% if a single sample is analyzed [48,49], this technique has been considered the gold standard for the diagnosis of Giardia in dogs and cats [39]. However, due to the intermittent excretion of cysts and because only one sample was collected rather than the three recommended [49] the occurrence obtained in this study could be underestimated”

Fernandes de Mendoça, F.; Pitella, A.; Emmerick, S.; Pereira, N. Assesment of the diagnostic performances of four methods for detection of Giardia duodenalis in fecal samplesfrom human, canineand feline carriers. J Microbiol Methods, 2018, 145, 73-78, doi:10.1016/j

Decock, C.; Cadiergues, M.C.; Larcher, M.; Vermot, S.; Franc, M. Comparison of Two Techniques for Diagnosis of Giardiasis in Dogs. Parasite2003, 10, 69–72, doi:10.1051/parasite/2003101p69.

Immunofluorescence assays (IFAs) can be used to stain fecal smears without initial concentration and has more sensitivity   than concentration/microscopy combination. 

I recommend reanalyzing the samples using IFA, previously to PCR assays.

Although we agree with the reviewer that IFA could have been use, unfortunately we did not keep the samples. However as mentioned before the zinc sulfate technique with the centrifugation method is the recommended diagnosis test for Giardia in dogs and cats

In results section

- Where obtained the positive control? 

The positive control for the PCR was obtained from the epidemiologic diagnosis and reference institute of Mexico (InDRE).

- Authors should include controls for Assemblages A, B and F in PCR, at least. It is not clear to the reader that the samples are Assemblages A.

The purpose of this paper was to find the possible zoonotic genotypes and being assemblage A the most common, we decided to use only that one. Although we agree with the reviewer that using both zoonotic genotypes (A, B) would have been helpful.

Reviewer 2 Report

Dear Authors

Concerning your manuscript animals-2223531-peer-review-v1 “Prevalence of Giardia intestinalis in cats from Queretaro and the risk to public health”, I believe it is an interesting and a relevant field of clinical research concerning cat parasites and their epidemiological surveillance and control on a local and worldwide basis. Besides, since the number of recent case reports regarding Giardia intestinalis in cats is growing at global level, being this manuscript the first study of this type in cats in Mexico, effort should be taken to develop more and more often the kind of observational cross-sectional study as reported here, above all to assess different clinical pictures and genotypes of Giardia sp. in domestic felids. And this type of original clinical research must have more visibility, since its main conclusions can drive veterinarians, animal shelter workers and cat owners for a more comprehensive work together towards a more assertive way of predicting the onset of this parasitic disease, not mentioning its repercussion on Public Health taken its zoonotic character.

Besides what will be pointed out, namely that your manuscript has potential to be published, the final decision on the publication of your manuscript at Animals, depends on the Editor final statement.

Regarding my reviews and comments, they are as follows:

Key-words

Page 1

Line 49 – Remove the numbers after each key-word and I also would include, if possible, the word Mexico.

1. Introduction

Page 2

Line 55 – Instead of Giardiasis you should write Giardiosis, because according to the paper T. KASSAI, M. CORDERO DEL CAMPILLO, J. EUZEBY, S. GAAFAR, Th. HIEPE & C.A. HIMONAS (1988) Standardized Nomenclature of Animal Parasitic Diseases (SNOAPAD), Veterinary Parasitology, 29 (1988) 299-326 pp., the name of animal parasitic diseases should use the suffix osis due to their chronic evolution, although human parasitologists prefer iasis. And this change should be performed along the text.

Line 78 – Instead of “symptoms”, write “clinical signs” since it is an expression more often used in veterinary medicine, once the animals can’t tell us their symptoms, rather we have to extract the clinical signs expressed by the animal. Replace that word in other parts of the text.

Line 96 – Instead of “…cats. [16,17].”, write “…cats [16,17].”.  

2. Materials and Methods

Page 3

Line 126 – Explain better the location of the town where the study was performed. The city of Santiago de Queretaro is in the north, close to sea, etc.   

Results

Page 5

Lines 211-212 – You mention the male and female prevalence are similar, both showing 24%, but on table 1 there is discrepancy since males had 24% and females 25% prevalence. Use the most correct one on the table and along the text.

Discussion

Page 7

Line 283 – Instead of “…results [51], Pooling…”, write “…results [51]. Pooling…”.

Funding

Page 8

Table A1 – Title – Lines 351-352. This title should be just before the table, meaning that they must be in the same page. The scientific name G. intestinalis should be in italic.

Table A2 – Title – Lines 365-366. The scientific name G. intestinalis should be in italic.

References

Pages 9-12

Check if all the scientific names are well written (first letter of genus with capital letter, while the first letter of species name must be written with lowercase) and in italic.

Best regards and good luck with your amendments.

Reviewer

Author Response

We appreciate the time that reviewer 2 took to enrich this work. Fortunately, we have been able to modify all the suggestions that this reviewer provided to improve the manuscript.

Concerning your manuscript animals-2223531-peer-review-v1 “Prevalence of Giardia intestinalis in cats from Queretaro and the risk to public health”, I believe it is an interesting and a relevant field of clinical research concerning cat parasites and their epidemiological surveillance and control on a local and worldwide basis. Besides, since the number of recent case reports regarding Giardia intestinalis in cats is growing at global level, being this manuscript the first study of this type in cats in Mexico, effort should be taken to develop more and more often the kind of observational cross-sectional study as reported here, above all to assess different clinical pictures and genotypes of Giardia sp. in domestic felids. And this type of original clinical research must have more visibility, since its main conclusions can drive veterinarians, animal shelter workers and cat owners for a more comprehensive work together towards a more assertive way of predicting the onset of this parasitic disease, not mentioning its repercussion on Public Health taken its zoonotic character.

Concerning your manuscript animals-2223531-peer-review-v1 “Prevalence of Giardia intestinalis in cats from Queretaro and the risk to public health”, I believe it is an interesting and a relevant field of clinical research concerning cat parasites and their epidemiological surveillance and control on a local and worldwide basis. Besides, since the number of recent case reports regarding Giardia intestinalis in cats is growing at global level, being this manuscript the first study of this type in cats in Mexico, effort should be taken to develop more and more often the kind of observational cross-sectional study as reported here, above all to assess different clinical pictures and genotypes of Giardia sp. in domestic felids. And this type of original clinical research must have more visibility, since its main conclusions can drive veterinarians, animal shelter workers and cat owners for a more comprehensive work together towards a more assertive way of predicting the onset of this parasitic disease, not mentioning its repercussion on Public Health taken its zoonotic character. 

Besides what will be pointed out, namely that your manuscript has potential to be published, the final decision on the publication of your manuscript at Animals, depends on the Editor final statement. 

Regarding my reviews and comments, they are as follows:

Page 1

Line 49 – Remove the numbers after each key-word and I also would include, if possible, the word Mexico.

The correction was made

Line 55 – Instead of Giardiasis you should write Giardiosis, because according to the paper T. KASSAI, M. CORDERO DEL CAMPILLO, J. EUZEBY, S. GAAFAR, Th. HIEPE & C.A. HIMONAS (1988) Standardized Nomenclature of Animal Parasitic Diseases (SNOAPAD), Veterinary Parasitology, 29 (1988) 299-326 pp., the name of animal parasitic diseases should use the suffix osis due to their chronic evolution, although human parasitologists prefer iasis. And this change should be performed along the text.

We agree with the reviewer, the terminology was changed as suggested

Line 78 – Instead of “symptoms”, write “clinical signs” since it is an expression more often used in veterinary medicine, once the animals can’t tell us their symptoms, rather we have to extract the clinical signs expressed by the animal. Replace that word in other parts of the text.

We totally agree, the correction was made

Line 96 – Instead of “…cats. [16,17].”, write “…cats [16,17].”

The correction was made

Line 126 – Explain better the location of the town where the study was performed. The city of Santiago de Queretaro is in the north, close to sea, etc.   

We agree with the reviewer, we provided information about the location in the paragraph. Lines [233-236]

Lines 211-212 – You mention the male and female prevalence are similar, both showing 24%, but on table 1 there is discrepancy since males had 24% and females 25% prevalence. Use the most correct one on the table and along the text.

The correction was made

Line 283 – Instead of “…results [51], Pooling…”, write “…results [51]. Pooling…”.

The correction was made

Table A1 – Title – Lines 351-352. This title should be just before the table, meaning that they must be in the same page. The scientific name G. intestinalis should be in italic.

Table A2 – Title – Lines 365-366. The scientific name G. intestinalis should be in italic.

We totally agree with the reviewer, the correction was made

Check if all the scientific names are well written (first letter of genus with capital letter, while the first letter of species name must be written with lowercase) and in italic.

The correction was made throughout the manuscript.

Reviewer 3 Report

Overall, this is an interesting study about the importance of Giardia intestinalis prevalence. My comments below indicate things that need to be revised or explained,

 1. Is the maker of fig2 wrong? I'm not sure about the size of the fragments in the figure2 according to the maker, but the size of the fragments is obviously not 70,100, and190 bp.

 2. Line211, the detection of G. intestinalis was similar in males and females (24% vs 24%). However, the results of table 1 shows that detection of G. intestinalis was 24% vs 25%. In addition, different description appear in line 257, “differences in prevalence were seen in each of the categories defined, such……”. It is contradictory that such descriptions appear, which is the real result?

 3. For genotyping, RFLP analysis showed that the 70 bp, 100 bp and 190 bp fragment patterns were considered to be genotype AI, and the 70 bp and 210 bp fragment patterns were considered to be the AII genotype. As shown in Figure 2, 210 bp band is similar to 190 bp band, and the latter is difficult to distinguish. How do you tell if the samples of 2, lane 5 and lane 6 are not a mix of genotype AI and genotype AII? I don't think you can tell if the pool is AII or a mix of AII and AII based on this approach, such as 2, 5, and 6 pools.

Author Response

We really appreciate the time reviewer 3 expend in reviewing our manuscript, we found his comments very appropriated and important to improve the quality of our paper.

Overall, this is an interesting study about the importance of Giardia intestinalis prevalence. My comments below indicate things that need to be revised or explained, 

  1. Is the maker of fig2 wrong? I'm not sure about the size of the fragments in the figure2 according to the maker, but the size of the fragments is obviously not 70,100, and190 bp.

We totally agree with the reviewer and the correction was made

  1. Line211, the detection ofG. intestinalis was similar in males and females (24% vs 24%). However, the results of table 1 shows that detection of G. intestinalis was 24% vs 25%. In addition, different description appear in line 257, “differences in prevalence were seen in each of the categories defined, such……”. It is contradictory that such descriptions appear, which is the real result?

We agree with the reviewer and the correction was made

  1. For genotyping, RFLP analysis showed that the 70 bp, 100 bp and 190 bp fragment patterns were considered to be genotype AI, and the 70 bp and 210 bp fragment patterns were considered to be the AII genotype. As shown in Figure 2, 210 bp band is similar to 190 bp band, and the latter is difficult to distinguish. How do you tell if the samples of 2, lane 5 and lane 6 are not a mix of genotype AI and genotype AII? I don't think you can tell if the pool is AII or a mix of AII and AII based on this approach, such as 2, 5, and 6 pools.

We agree with the reviewer, we were not able to show different patterns between 210 and 190 pb. This could be explain by the possibility that those pools were a mixed of assemblages A1 and A2 (lines 2, 5 and 6), due to the fact, that the samples were pools from different positive animals. However, these results don’t change the purpose of the study which was to find the occurrence of Giardia in cats and the presence of zoonotic assemblages (A1, A2). The absence of the 100 pb pattern band on lines 1, 3, 4 and 7 indicates the presence of assemblage A2

Reviewer 4 Report

According to the authors, the objective of this study was to determine the prevalence of this parasite in the feline population of the city of Santiago de Querétaro, Mexico, and to identify the genotypes present to determine the role played by this host in public health, this being the first study of this type to be carried out in the country.

Although it is a simple study that basically determines the presence of Giardia duodenalis as well as the molecular variability, this work provides knowledge on the presence in animal species that are pets and that have a relevant role in the transmission of zoonotic pathogens due to their close animal-human relationship.

The paper is clear, well written and well presented; however, there are some issues that should be considered.

Title

 It is only a matter of opinion and according to Thompson and Monis (Thompson RC, Monis PT. Variation in Giardia: implications for taxonomy and epidemiology. Adv Parasitol. 2004; 58:69–137. doi: 10.1016/S0065-308X(04)58002-8) the most appropriate way to name this species of Giardia is Giardia duodenalis. Replace throughout the manuscript

On the other hand, the term “prevalence” would refer to the entire cat population, so they suggested substituting the term prevalence for “presence” or “occurrence”. Replace throughout the manuscript

Simple summary

Line 28: transmitter is not an appropriate terminology in this context; I suggest replacing it with reservoir, which also implies the transmission of pathogens to other individuals.

Lines 30-32: Giardia duodenalis is actually a cryptic species complex consisting of eight (A-H) lineages called assemblages with marked differences in host range and host specificity. Please substitute genotypes, genotyping, etc., for assemblages throughout the manuscript

Introduction

Lines 52-53: Giardia duodenalis is one of the most common gastrointestinal parasites that affects humans, but also to other animal species. Please modify this, and begin the paragraph with the full scientific name, without abbreviation.

Lines 98-107: This information could be included in a summary table, incorporating a small review of the data on the presence of Giardia in cats worldwide; this will reduce the introduction and allow the data to be more comparable for the discussion.

Materials and Methods

Study setting and sampling

Lines 127-129: Here it is said that one origin of the cats comes from stray or ownerless cats from the municipal animal control unit (UCAM), however, in the simple summary it is detailed that they come from shelters, please modify this.

Sample pooling

Lines 146-148: Please explain better how the groups are made, how many samples are included in each group, since the way it is described in this version is not well understood.

PCR

It is not necessary to detail the pcr conditions if they are those described in Caccio et al (38). I suggest combining PCR, RFLP and sequencing in the same section.

Results

Sequencing

Why are these two sequences not deposited in GenBank? Once the sequences are deposited in GenBank and with the accession number indicated in the manuscript, it would not be necessary to provide the data in Appendix A.

Author Response

We really appreciate the time reviewer 4 expend in reviewing our manuscript, we found his comments very appropriated and important to improve the quality of our paper  

According to the authors, the objective of this study was to determine the prevalence of this parasite in the feline population of the city of Santiago de Querétaro, Mexico, and to identify the genotypes present to determine the role played by this host in public health, this being the first study of this type to be carried out in the country.

Although it is a simple study that basically determines the presence of Giardia duodenalis as well as the molecular variability, this work provides knowledge on the presence in animal species that are pets and that have a relevant role in the transmission of zoonotic pathogens due to their close animal-human relationship.

The paper is clear, well written and well presented; however, there are some issues that should be considered.

.

It is only a matter of opinion and according to Thompson and Monis (Thompson RC, Monis PT. Variation in Giardia: implications for taxonomy and epidemiology. Adv Parasitol. 2004; 58:69–137. doi: 10.1016/S0065-308X(04)58002-8) the most appropriate way to name this species of Giardia is Giardia duodenalis. Replace throughout the manuscript

On the other hand, the term “prevalence” would refer to the entire cat population, so they suggested substituting the term prevalence for “presence” or “occurrence”. Replace throughout the manuscript

We agree with the reviewer, the corrections were made throughout the manuscript.

Line 28: transmitter is not an appropriate terminology in this context; I suggest replacing it with reservoir, which also implies the transmission of pathogens to other individuals.

We agree with the reviewer, the terminology was changed as suggested.

Lines 30-32: Giardia duodenalis is actually a cryptic species complex consisting of eight (A-H) lineages called assemblages with marked differences in host range and host specificity. Please substitute genotypes, genotyping, etc., for assemblages throughout the manuscript

We agree with the reviewer, the corrections were made throughout the manuscript.

Lines 52-53: Giardia duodenalis is one of the most common gastrointestinal parasites that affects humans, but also to other animal species. Please modify this, and begin the paragraph with the full scientific name, without abbreviation.

The correction was made

Lines 98-107: This information could be included in a summary table, incorporating a small review of the data on the presence of Giardia in cats worldwide; this will reduce the introduction and allow the data to be more comparable for the discussion.

We agree with the reviewer, a table was incorporated to present the prevalence in cats around the world. Line 181

Lines 127-129: Here it is said that one origin of the cats comes from stray or ownerless cats from the municipal animal control unit (UCAM), however, in the simple summary it is detailed that they come from shelters, please modify this.

The correction was made

Lines 146-148: Please explain better how the groups are made, how many samples are included in each group, since the way it is described in this version is not well understood.

This was addresses on lines 256-260

It is not necessary to detail the pcr conditions if they are those described in Caccio et al (38). I suggest combining PCR, RFLP and sequencing in the same section.

We totally agree with the reviewer, the PCR conditions were deleted as suggested

Why are these two sequences not deposited in GenBank? Once the sequences are deposited in GenBank and with the accession number indicated in the manuscript, it would not be necessary to provide the data in Appendix A.

The sequences were not deposited in the GeneBank because they are not new sequences having a query cover of 100% with other reference sequences that are in the GenBank.

Round 2

Reviewer 1 Report

The objective of this study was to determine the prevalence of this parasite in the feline population, and to identify the genotypes present to determine the role played by this host in public health. However, the study does not include the analysis of the two zoonotic genovarieties, only assemblage A. The authors indicate that assemblage A is the most frequent, but there are studies that also demonstrate the presence of assemblage B

-Karimi P, Shafaghi-Sisi S, Meamar AR, Razmjou E. Molecular identification of Cryptosporidium, Giardia, and Blastocystis from stray and household cats and cat owners in Tehran, Iran. Sci Rep. 2023 Jan 27;13(1):1554. doi: 10.1038/s41598-023-28768-w. PMID: 36707690; PMCID: PMC9883249.

 -Sursal N, Simsek E, Yildiz K. Feline giardiasis in Turkey: prevalence and genetic and haplotype diversity of Giardia duodenalis based on the β-giardin gene sequence in symptomatic cats. J Parasitol. 2020 Oct 1;106(5):699-706. doi: 10.1645/19-183. PMID: 33120408.

 It is especially interesting to know the presence of both assemblages given the high prevalence found in cats with owner. Therefore, I consider that the study should be completed as I indicated in the first review.

Author Response

First of all, we are very thankful to the reviewer for the time he/she took to review this paper and for the very valuable comments

We agree with the reviewer, however, at this point it is impossible for us to do the assemblage B due to the fact that we used all the DNA to do the study. We never thought to work with the assemblage B because of the fact that the presence in Mexico is very low and has only been found twice in humans and never in domestic animals.

Reviewer 3 Report

This is a very interesting and meaningful work. The author answered or corrected the previous review comments. There may be some details to deal with. The results of PCR in Figures 1 and 2 may need to be cut (trim white space in the figures) to be makes the results easier to read. I think this article can be published after correction.

Author Response

We believe that the correction was made

Reviewer 4 Report

Thanks to the authors for answering all the questions raised and modifying the manuscript accordingly.

However, there are still two issues that remain unclear to me.

The first is on the grouping of positive samples into pools. The occurrence of Giardia was detected in 50 samples of the total, the authors prepared 7 pools of 4 positive samples each, however with this grouping we would have a total of 28 samples, there would be another 22 samples to be pooled. Could you please clarify this point?

The other question is about the sequences generated, once again I suggest that these sequences be deposited in GenBank, as it is very useful for future studies, and also they would be the first sequences of Giardia duodenalis deposited in GenBank from Mexican cats, this is useful when making phylogenetic trees with samples from different geographical origins.

Author Response

First of all, we are very thankful to the reviewer for the time he/she took to review this paper and for the very valuable comments

We were unable to make more pools because of the low volume of sample obtained from the cats, all the positive samples which had enough amount of faeces were pooled.

The sequences were submitted to the GenBank, but it will take some days until they are shown.  submission ID is: 2682558